# Validation of the StimQ$_2$: A parent-report measure of cognitive stimulation in the home

**Carolyn Brockmeyer Cates**[1☖‡*], **Erin Roby**[2☖‡*], **Caitlin F. Canfield**[2],
**Matthew Johnson**[3], **Caroline Raak**[4], **Adriana Weisleder**[5], **Benard P. Dreyer**[2], **Alan L. Mendelsohn**[2]

**1** Department of Psychology, Purchase College, State University of New York, Purchase, New York, United States of America, **2** Department of Pediatrics, Division of Developmental and Behavioral Pediatrics, NYU Grossman School of Medicine, New York, New York, United States of America, **3** Educational Testing Services, Princeton, New Jersey, United States of America, **4** Ferkauf Graduate School of Psychology, Yeshiva University, New York, New York, United States of America, **5** Department of Communication Sciences and Disorders, Northwestern University, Chicago, Illinois, United States of America

☖ These authors contributed equally to this work.
‡ These authors shared first authorship on this work.
* Carolyn.cates@purchase.edu (CBC); erin.roby@nyulangone.org (ER)

**Data Availability Statement:** The data is available at: https://nyu.databrary.org/volume/1617.

**Funding:** This research was supported by the National Institutes of Health (R01 HD047740 01-

## Abstract

Considerable evidence demonstrates the importance of the cognitive home environment in supporting children's language, cognition, and school readiness more broadly. This is particularly important for children from low-income backgrounds, as cognitive stimulation is a key area of resilience that mediates the impact of poverty on child development. Researchers and clinicians have therefore highlighted the need to quantify cognitive stimulation; however existing methodological approaches frequently utilize home visits and/or labor-intensive observations and coding. Here, we examined the reliability and validity of the StimQ$_2$, a parent-report measure of the cognitive home environment that can be delivered efficiently and at low cost. StimQ$_2$ improves upon earlier versions of the instrument by removing outdated items, assessing additional domains of cognitive stimulation and providing new scoring systems. Findings suggest that the StimQ$_2$ is a reliable and valid measure of the cognitive home environment for children from infancy through the preschool period.

## Introduction

There is longstanding evidence demonstrating that the early home environment is associated with burgeoning developmental capacities and early academic achievement [1]. In addition to factors such as parental warmth and responsivity contributing to social-emotional health and well-being [2], cognitive stimulation in the home has also been highlighted as a key factor related to children's developing cognitive, language, and social development [3–5]. For example, cognitive stimulation in the home, including availability of learning materials to support play and reading, parental support for learning new skills, overall parental verbal responsivity, and engagement in shared bookreading have been shown to be related to enhanced cognitive, linguistic, and social-emotional skills in young children, with cascading impacts on school preparedness and later achievement.

09, Supplement 3R01HD047740-08S1) awarded to ALM. The funders had no role in study design, data collection and analysis, decision to publish, or preparation of the manuscript.

**Competing interests:** The authors have declared that no competing interests exist.

Quantifying cognitive stimulation in the home is thus important for all children but may be of particular importance for children from families with low-socioeconomic status (SES). Longstanding systemic injustices and structural racism have resulted in reduced opportunities for children who are growing up in poverty, a disproportionate number of whom are Black and Latinx, to be afforded the resources and experiences needed to prepare for the demands of formal education [6–8]. Over fifty years of research has shown that these inequities have significant negative effects on children's health and development, particularly with regards to early language and cognitive skills associated with school readiness [9]. Especially troubling is the fact that these early disparities can persist into elementary school, high school, and beyond, ultimately leading to decreased educational achievement, career advancement, economic stability, and health in adult life [10–13]. Developing interventions to counteract these poverty-related disparities has thus been the major focus of child development studies over the past several decades, and is now considered to be a significant public health issue in the United States [14–16].

While poverty-related disparities in school readiness and academic achievement are multifactorial, cognitive stimulation has emerged as a primary target of many interventions aiming to enhance outcomes for children from low-SES backgrounds [17–20]. This is both because cognitive stimulation has been identified and highlighted as a positive experience in childhood that can buffer the negative consequences of poverty [15, 21, 22], and because it is demonstrated to be a modifiable factor [23–26]. Therefore, the capacity to measure cognitive stimulation in the home is a pressing concern not just for developmental research, but also for identification of family strengths and challenges and for measurement of intervention efficacy.

Historically, the most commonly used modalities for assessing cognitive stimulation in the home have been resource-intensive, often requiring observation and coding/interpretation by trained professionals in home- and lab/clinic-based settings. To address the need for a valid and reliable method for assessing the cognitive home environment in a less resource-intensive way, Dreyer and colleagues [27] developed the StimQ, an office-based assessment of children's cognitive stimulation in the home (including parent verbal responsivity, parent engagement in developmental advance, reading behaviors, and availability of learning materials). Prior studies of the original StimQ have demonstrated good internal and external reliability for both an infant version for use from 5–12 months of age (StimQ-Infant, α = .85) and a toddler version for use from child age 12–36 months (StimQ-Toddler, α = .83), as well as construct, criterion-related, and predictive validity [27, 28]. Given evidence that cognitive stimulation in the home during the preschool period continues to play an important role in child school readiness outcomes [29, 30], a preschool version of the StimQ was also created for children aged 36 through 72 months of age, with validity and reliability preliminarily established [31]. Since its development, the StimQ has been used in part or in its entirety in at least 93 original published research studies [32–34], including 29 from our lab [23, 35–37], and 16 from countries outside the United States [38–40], demonstrating continued interest in and need for this instrument. In addition, while the original StimQ was developed and validated in both English and Spanish, it has since been translated into a number of other languages including Chinese, Dutch, French, Italian, Portuguese, Thai, and Turkish [41–44]. However, since the StimQ was developed nearly 25 years ago, adjustments may be needed to optimize its use for the present time. In this study, we sought to revalidate the instrument to address concerns regarding appropriateness of certain items (particularly in light of technological advances in children's games and media) and expand its flexibility and ease of implementation. Furthermore, as variability in psychometric characteristics across forms and scales suggested the possibility of improvement, we aimed to modify existing forms to optimize length of administration through addition and/

or removal of items and to develop smaller components that could be used individually to fit the needs of the researcher(s).

## Existing methods for measuring cognitive stimulation in the home

**Observational strategies.**   While multiple modalities have been used to measure cognitive stimulation in the home, each has distinct advantages and disadvantages. Observational instruments, administered in both homes and lab- or clinic-based settings, are often considered the gold standard in research aimed at capturing children's naturalistic environments and interactions [5, 45]. Several existing instruments have demonstrated good reliability and validity including The Home Observation for Measurement of the Environment (HOME) Inventory, which provides an observable snapshot of the child's home setting and experiences (interrater reliability 90–95%; $\alpha$ = .44 to .89) [5, 46–48] and the Homelife Interview, which assesses parental responsiveness, provision of learning activities, parental supervision, parental communication skills, and routines [49]. These measures have been shown to be predictive of a wide variety of child outcomes [47, 49–53]. Direct observation measures are especially useful for capturing information about children's early literacy exposure and experiences, access to learning materials (including toys that facilitate symbolic play), and naturalistic or spontaneous parent-child interactions and exchanges. While home observation confers unique strengths for capturing information about cognitive stimulation in the home in its naturally occurring environment, there remain significant limitations to this methodology- most notably, its resource-intensive nature. Substantial investment of funds, time, training of highly skilled staff, and subsequent data analyses are required to conduct these observations. Additionally, access to individuals' homes must be granted; this poses a significant challenge and limitation, as some families may not feel comfortable letting researchers into their home. This introduces a potential bias, in that families reached using this method may be those more willing to grant visitors access to their homes.

Observational methods have also been designed for use in lab- and clinic-based settings, such as the Pediatric Review and Observation of Children's Environmental Support and Stimulation (PROCESS), which was designed to assesses parent-child interaction during a health supervision visit and is conducted directly by the child's pediatrician (correlation with HOME and home observation, $r$ = .34 to .67) [54]. While posing a viable alternative to observation in the home, there remain some challenges with this methodology. First, there are significant costs related to the time and training needed to use this methodology. Second, although observations in both home and lab/clinic settings may be particularly useful for assessing qualitative features of parent-child interactions, they provide only a snapshot of these behaviors, and are less able to inform on the occurrence of cognitively stimulating interactions (e.g., talking, reading, and playing) in everyday life. Moreover, the nature of these observations may be impacted by caregivers' knowledge that they are being observed and/or recorded [5]. The knowledge of being observed has been shown to lead to intentional or unintentional changes in the usual behaviors of those being observed, which can be motivated by social desirability [55, 56].

**Parent-report.**   Another way of assessing children's cognitive home environment is through parental report. Several parent surveys have been developed and have become a popular tool in measuring parenting practices and parent-child shared/joint interactions in the home. These surveys range in complexity, from documenting whether or not an activity is done (i.e. yes or no questions) to determining relative frequency (e.g., a frequency Likert scale within a specific time period, like "last week").

Some examples of these parent report surveys include the Parent Reading Belief Inventory (PRBI) [57–59], the Stony Brook Family Reading Survey (SBFRS) [60], the Home Literary

Activities Questionnaire (HLAQ) [61, 62], Home Screening Questionnaire (HSQ) [63, 64], and the SharePR [65]. In addition to ranging in complexity, these instruments often narrowly target specific aspects of cognitive stimulation, such as the home literacy environment. For instance, maternal literacy, child's interests in reading, exposure to joint reading, maternal beliefs about reading aloud, and frequency of shared reading are some of questions addressed by the FS, the PRBI, the SBFRS, and the HLAQ.

Furthermore, while such surveys are generally easy to complete and provide important and varied information about a single aspect of cognitive stimulation in the home (such as home literacy environment), there is a need for a measure that more comprehensively assesses cognitive stimulation. For example, it is also critical to measure other dimensions of cognitive stimulation beyond reading because additional factors such as parent verbal input/responsivity, engagement in play, and teaching activities have also been shown to support children's school readiness abilities [3]. The inclusion of these behaviors also provides the opportunity to capture variation in strengths of different families and cultural groups. Because of this, other measures, including the StimQ, have attempted to capture a more holistic picture of the cognitive home environment. There is also a growing need for a well-validated instrument that broadly assesses parent-child interactions and is specifically designed to measure aspects of parenting across ethnically diverse parents with low income and with infants and young children 0 to 6 years. Latinx families represent a particularly important population of focus because despite being the largest ethnic minority group in the United States (~62 million) [66], they have been largely understudied in this area. In addition, although there is broad within-group heterogeneity, poverty and racism in the U.S. are strongly linked [67], and growing up in families with low incomes is much more common for Latinx (53%) children compared with White children (26%), ultimately resulting in disparities in achievement in math and reading [68].

While parent-report/survey measures avoid the resource intensity of home or laboratory observations, they continue to be vulnerable to the social desirability bias. As many of the behaviors targeted by these surveys are activities that U.S. society values as markers of "good parenting", parents may (intentionally or unintentionally) respond based on how they think they are expected to act in their role as parents. In addition to inflating scores, this can also lead to bias in how these measures capture relevant behaviors across racial and ethnic groups, as parents from minoritized groups may have different expectations about the behaviors that constitute "good parenting" and may be less likely than those from majority groups to endorse activities that are valued in the dominant culture. Therefore, a measure of cognitive home environment is needed that aims to reduce this bias as much as possible, a key consideration of the StimQ.

**StimQ.** The StimQ is an interviewer-administered questionnaire that can be administered in a variety of settings, including clinics, childcare/school, and research laboratories and can be downloaded freely online. It does not require observation or video coding, which significantly decreases the amount of time needed for both administration and training. It takes approximately 20 minutes to complete the full StimQ, which includes four main subscales—Availability of Learning Materials (ALM; variety of toys), Reading (READ; books / reading activities), Parental Involvement in Developmental Advance (PIDA; teaching activities) and Parental Verbal Responsivity (PVR; verbal interactions), with continuous subscale scores for each of these domains. It is available in English and Spanish, had been translated and/or adapted across a number of different languages.

The StimQ was designed to minimize social desirability bias by requiring that any responses endorsed by the parent be followed with requests for additional information, such as specific examples of that behavior. This is done to increase the likelihood that "Yes" responses reflect true engagement in the activity, and to encourage "No" responses when the activity is not

performed (because it will be difficult for the parent to come up with an example or additional information). For instance, one item on the Infant StimQ PVR scale asks, "Do you usually talk to your baby while you are feeding her and tell her about what is going on, or is she too young to talk with yet?" If the parent responds in the affirmative, they are then asked, "Can you give me some specific examples of these conversations?" Credit for the item is only given if parents can give an example of a circumstance in which they spoke to their baby while feeding them. Data for the original StimQ were collected at a large, urban public hospital (a small percentage were collected at private pediatric clinics) through a series of studies focused on mothers of children between 12 to 36 months. Mothers were primarily Latinx or Black immigrants with low socioeconomic status. A smaller percentage were White with medium to high SES. The survey was administered to mothers by trained interviewers in their primary language during their pediatric visit. The original StimQ demonstrated good internal and external reliability, as well as good construct, criterion-related, and predictive validity (r = .55 for correlation with the HOME Inventory) [27].

**StimQ$_2$: The revision process.** In this paper, we aim to re-validate the StimQ and introduce a revised version, the StimQ$_2$, which was developed to meet the need for an updated reliable and valid survey comprehensively measuring cognitive stimulation in the home in a non-resource-intensive manner. While the original StimQ forms had adequate reliability and validity, variability in psychometric characteristics of this instrument suggested the possibility for improvement through addition and/or removal of items. Over the course of years that the original StimQ had been used with families, it became evident that some of the items, especially on the ALM scale, required revision/removal due to technological advances (e.g., whether parents owned a "children's record player/tape player"). These changes were informed, in part, by informal feedback from parents regarding the kinds of materials they did or did not own. Additionally, research findings emerging since the development of the original instrument illuminated that the original StimQ underrepresented some of the critical aspects of cognitive stimulation in the home (e.g., reading quality, promotion of early pretend play, and promotion of early self-regulation) that have been linked with enhanced developmental outcomes and are relevant across a broader range of cultural traditions [69–71]. Therefore, using an established evidence-base of parenting practices related to early developmental outcomes, new items were added to the revised StimQ instrument to more adequately address these domains of cognitive stimulation in the home. Modifications to questions were also made using feedback from bilingual (English and Spanish), bicultural members of our team with experience administering the StimQ to families in order to address language and cultural differences. Finally, feedback from investigators using the StimQ both from our lab and beyond indicated appeal and benefit for researchers to be able to select subscales of the StimQ rather than administering the full instrument in all cases. The StimQ$_2$ was designed to include the original four subscales (ALM, READ, PIDA, and PVR), each comprised of specific components (e.g., READ bookreading quality, diversity concepts/content, bookreading quality), which could be validly and reliably used alone or together with the full instrument. This same structure exists for the infant, toddler, and preschool forms. We also aimed to validate two scoring systems for the revised scale: 1) one with all four subscale scores combined together (StimQ$_2$-Total), and 2) a "core" cognitive stimulation score combining the scores from the READ, PIDA, and PVR subscales (StimQ$_2$-Core) but excluding ALM (see S1 Appendix). This was done to account for the fact that the ALM subscale takes the most time to administer. In this study, we therefore aimed to: 1) improve the psychometric characteristics of the StimQ using item-response theory (IRT) analysis, and 2) to test the concurrent validity of this revised instrument through analyses of data collected from families living in low-income, urban communities.

## Methods

### Study design

Cross-sectional analyses were conducted using data from a larger study of a pediatric primary care intervention promoting child development and responsive parenting—The Bellevue Project for Early Language, Literacy, and Education Success (The BELLE Project) [17, 23, 26]. Mother-infant dyads were enrolled during their postpartum stay in the level I newborn nursery of a large urban public hospital between November 2005, and October 2008. Inclusion criteria included normal singleton birth (i.e. weight, full term gestation) no significant medical or developmental complications, and planned pediatric care at the institution. Mothers had to be of legal age ($\geq$ 18 years), the child's legal guardian, and speak English or Spanish.

### Participants

Mother-child dyads who were assessed on at least one of five possible time points (child age 6, 14, 24, 36, and 54 months) and had data for the variables of interest were included in analyses ($N$ = 546). Mothers did not have to maintain participation across all time points to be included. The total number of mothers enrolled in the larger trial was 675, with up to 675 available for follow up at 6, 14, and 24 months and up to 450 available for follow up at 36 and 54 months). Participants in the current study include 407 at child age 6 months (60% of those available for follow up; $M$ = 6.87; $SD$ = 1.25), 324 at child age 14 months (48%; $M$ = 15.49; $SD$ = 1.57), 374 at child age 24 months (55%; $M$ = 25.69; $SD$ = 2.32), 302 at child age 36 months (67%; $M$ = 39.12; $SD$ = 3.65), and 273 at child age 54 months (61%; $M$ = 57.93; $SD$ = 4.49). All mothers identified themselves as the child's primary caregiver. Parents were asked to report their race: 6% identified as Black, 1% identified as Asian, 7% identified as White, 12% identified as Native American, 1% identified as Native Hawaiian or Pacific Islander and 75% chose "other race". Overall, 93% of families identified as Hispanic/Latinx. Mothers' were interviewed in either Spanish (76%) or English (24%). Most mothers were born outside the United States (87%), and emigrated from 30 different countries. A majority reported being married/living with partner (84%), and 91% had low-socioeconomic status (SES) as defined using Hollingshead four factor index [72]. Despite primarily having low SES, mothers varied considerably in the number of years of education they had completed ($M$ = 10.01; $SD$ = 3.63; $R$ = 1 to 21). About 43% of primary caregivers had a high school education; 41% of children were first-born, and 51% were female. No significant differences were observed on demographic variables across time points. Institutional review board approval was obtained from NYU School of Medicine, and additional research approval was obtained from Bellevue Hospital Center, and New York City Health + Hospitals. Participating mothers provided written informed consent to participate in the study.

### Assessments

In addition to administering the StimQ to assess cognitive stimulation in the home, children's social, cognitive, and language development were assessed at each time point using a combination of parent-report and direct observation. This allowed for analyses of concurrent validity between the StimQ and critical domains of child development. Spanish/English bilingual research assistants administered all assessments. Surveys were conducted in parents' primary language. Direct assessments of children's abilities were administered in children's dominant language, which was assessed using the Simon Says subtest of the Pre-Language Assessment Scales [73].

## StimQ

The StimQ measure utilized included the original subscales (ALM, READ, PIDA, and PVR) with some pilot items added. The survey was administered to mothers by trained interviewers using age-appropriate scales at five time points: StimQ-Infant at 6 months, StimQ-Toddler at 14 and 24 months, and StimQ-Preschool at 36 and 54 months of child's age (for detailed information on StimQ administration procedures, see [27, 74].

## Assessment of child development: Infancy

**Infant communication.** Early communication was assessed at child age 6 months using four items from the "Emotion and eye gaze" subscale and two items from the "Communication" subscale of the Communication and Symbolic Behavior DP (CSBS DP) checklist, an assessment of predictors of language development such as eye gaze, gestures, understanding, and play. The CSBS DP is a parent-report instrument in which parents are asked to describe various aspects of their children's behavior. The "Emotion and eye gaze" subscale consists of questions such as "Does your child look at you while (s)he is happy?", answered on a three-point Likert scale (0–2) with 0 representing, "Never true of my child," and 2 representing, "Always true of my child". The items from the "Communication" subscale (e.g., "When you are not paying attention to your child, does (s)he try to get your attention?") were scored using the same Likert scale. The CSBS DP checklist has been shown to be valid for assessing children from diverse populations aged 6–24 months, and has good internal consistency (alpha = .96) and interrater reliability (alpha = .88-.89) [75].

**Infant temperament.** Infant temperament was assessed at 6 months using the Revised Infant Temperament Questionnaire Short Form (SITQ). The SITQ includes 7 items that measure four features of temperament including regularity, activity, intensity and soothability and includes questions such as, "The baby gets sleepy at about the same time each evening", "The baby moves a lot while lying awake". These questions are answered on a three-point Likert Scale (1–3) with 1 representing, "almost never", 2 representing "sometimes", and 3 representing, "often". The SITQ has demonstrated good internal consistency (domain α's = .57-.76) and test-retest reliability (.77-.90) [76].

## Assessment of child development: Toddlerhood

**Toddler cognitive ability.** The cognitive subtest of the Bayley Scales of Infant and Toddler Development, Third Edition (Bayley III) was used to assess cognitive development at 14 and 24 months (cognitive subtest, alpha = .91) [77]. The Bayley III is highly correlated with other measures of language and cognition such as the Bayley II, the Wechsler Preschool and Primary Scale of Intelligence, and the Preschool Language [78–80]. While the Bayley has been normed in English with efforts to avoid gender, racial/ethnic or socioeconomic biases, it has also been used extensively with Spanish-speaking and bilingual families [81–83].

**Toddler language ability.** Child language ability was measured at 14 and 24 months using Preschool Language Scale-4th edition [79], which was administered in either Spanish or English depending on the primary language spoken in the home as reported by the parent present. The PLS-4 is an observational measure comprised of two indicators: (1) Standard Expressive Communication Ability, and (2) Standard Auditory Comprehension. This test has been found to reliably assess language ability in children from birth through age 6 years 11 months (Cronbach's α = .93).

**Toddler social-emotional skills.** At 14 and 24 months, socioemotional skills were assessed via mother interviews using three subscales from the Infant-Toddler Social and Emotional Assessment-Revised (ITSEA): Imitation/Play, Attention, and Separation Distress. These

subscales were selected to assess key dimensions of socioemotional development at this age, including social skills, attention, and behavior. The ITSEA includes items describing behaviors that the parent rates as "not true/rarely," "somewhat true/sometimes," or "very true/often", with scores ranging from 0 to 2. It is available in English and Spanish and has been validated on children 12–36 months.

### Assessment of child development: Preschool

**Preschool cognitive ability.** As in the toddler period, The Bayley Scales of Infant and Toddler Development, Third Edition (Bayley III) was used to assess cognitive development at child age 36 months. At child age 54 months, two subtests of the Woodcock-Johnson III Tests of Cognitive Abilities/Bateria Woodcock- Muñoz Pruebas de habilidad Revisada were used to estimate child's cognitive abilities: (1) Visual Matching subtest, which measures the Catell-Horn-Carroll (CHC) processing speed factor (Gs) and (2) Non-Verbal Working memory [84, 85]. (Mather, 2001; The tests are normed in English (Woodcock-Johnson, validated on 6300 subjects) and in Spanish (Woodcock-Muñoz, standardized on 2,000 Spanish-speakers from Mexico, Puerto Rico, Costa Rica, Spain, Argentina, and Peru, as well as five US states).

**Preschool language ability.** At child age 36 months, language abilities were assessed using the Clinical Evaluation of Language Fundamentals Preschool-2 (CELF-2), which provides measures of expressive, receptive, and core language (Semel, Wiig, Secord, 2004). The CELF-2 is normed in English and Spanish for use with children between the ages of 3–6:11 years. Both the English and Spanish versions of the CELF have good test-retest reliability ranging from .91-.94, good internal consistency, >.90 for composite scores, and strong inter-rater reliability, .95-.98 [86].

At child age 54 months, expressive and receptive vocabulary were assessed using the Spanish-bilingual edition (SBE) of the Expressive One Word Picture Vocabulary Test (EOWPVT) and Receptive One Word Picture Vocabulary Test (ROWPVT). They are standardized for use with children between 4–12 years and has been validated for use with English and Spanish-speaking bilingual children. It provides a conceptual vocabulary score that captures Spanish-English bilingual children's vocabulary knowledge across both languages. The EOWPVT/ROWPVT-SBE has good internal reliability, α = .92-.98, and good test-retest reliability, correlation = .91-.92 [87, 88].

**Preschool behavior.** The Behavior Assessment System for Children was used to evaluate children's socio-emotional characteristics, specifically externalizing problems [89]. Parents responded to questions on the Externalizing Problems Scale, which gathers their perceptions on their child's: attention problems (e.g., Is easily distracted), social skills (e.g., Begins conversations appropriately), hyperactivity (e.g., Has poor self control), and aggression (e.g., Breaks other children's things) using a 4-point frequency Likert scale including, "never", "sometimes", "often", and "almost always". Both the English and Spanish versions have been shown to have strong sensitivity and specificity in diagnostic applications. It has high internal consistency (Cronbach's α coefficients > .80) and test-retest reliability [89].

### Data analytic plan

In order to improve the psychometric characteristics of the StimQ, we utilized item-response theory (IRT) analysis to determine which items from the original StimQ should be retained and to evaluate factor structure and internal consistency of the StimQ scale and subscales. We then performed reliability and subscore analyses of the StimQ subdomains to estimate the added value of considering these subdomains individually, in addition to the StimQ total score. Finally, we examined concurrent relationships between StimQ2 subscales and

components and child development outcomes in cognitive, language, and social-emotional domains. Missing data resulting from variation in timing of addition of pilot StimQ items were addressed through mean imputation. Data was imputed for at least one item at the following rates across time points: 6 months (15.76%), 14 months (17.65%), 24 months (.53%), 36 months (5.30%), 54 months (.73%).

## Results

### Results for Aim 1: Analyses of psychometric characteristics

**Selection.** Variability in psychometric characteristics of the original StimQ suggested the possibility for improvement through addition and/or removal of items. Item-response theory (IRT) analyses were conducted to determine retention or removal of items based on difficulty and discrimination parameters. Items were retained/removed to maintain balance in difficult and limit extremes ($|z| \geq 3$), and to maintain low discrimination ($< 0.5$) [90]. See S1 Table for details on the number of items that were dropped and added across the original and StimQ$_2$ versions.

**Factor structure and internal consistency.** To validate the internal structure of the StimQ$_2$ scale and subscales we fit multi-dimensional item response theory (MIRT) models for each age-level using the mirt package [91] in R (version 4.0.3) [92]. For each age-level we fit and compared four models of increasing complexity. First, we fit a unidimensional generalized partial credit model (GPCM) [93], which is equivalent to the two-parameter logistic (2PL) model when the item only has two response categories. This model assumes that all items load onto a single factor, and models the probability of responding in score category $k$ conditional on a latent variable $\theta$ that represents the individual's level on the single underlying construct (see Fig 1a). The parametric form of the model is:

$$P_{jk}(\theta) = \frac{\exp\left(ka_j\theta - d_{jk}\right)}{1 + \sum_{m=1}^{K_j} \exp\left(ma_j\theta - d_{jm}\right)},$$

where $K_j$ is the maximum response category for item $j$ and $d_{j0} = 0$ for all items $j$.

The second model (see Fig 1b) we examined was a simple structure multidimensional IRT (MIRT) model that assumed four dimensions corresponding to the subscales discussed in the

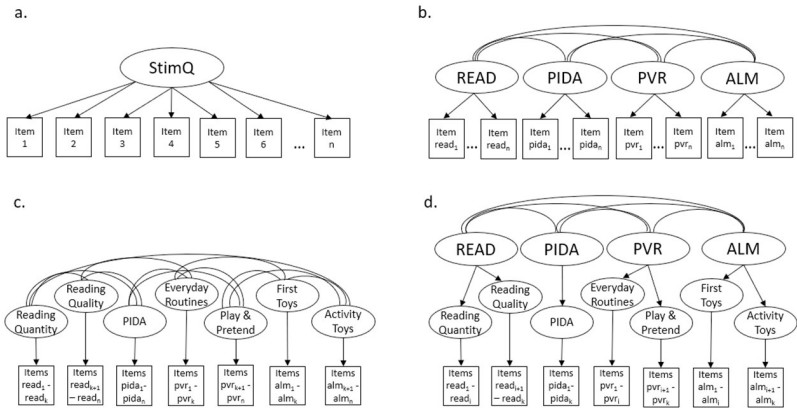

**Fig 1. Multidimensional item response theory (MIRT) models examining the structure of the StimQ$_2$.** Note: a. Unidimensional generalized partial credit model; b. Simple structure MIRT model; Component MIRT model; d. Multivariate bi-factor model. Not all subscales or components shown.

Introduction and shown in Table 1 (i.e., READ, PIDA, PVR, and ALM). The simple structure model assumes that each item measures a single dimension of the construct and that the four dimensions are correlated with one another. The underlying IRT model remains same as the GPCM above; however, instead of being a single latent variable, the underlying construct is now a vector of the four dimensions: $\theta = (\theta_1, \theta_2, \theta_3, \theta_4)$. The item response distribution, or slope, of each item now depends on the dimension of $\theta$ associated with that item, such that, if an item is highly discriminative within its dimension, it will have a greater influence on the overall outcome.

The third and fourth models (Fig 1c and 1d) we considered both take into account the components within each subscale of the StimQ$_2$ (READ, PIDA, PVR, and ALM) as described in the introduction and shown in Table 1. The third model treated each of the components as a separate dimension in the model, and therefore has 8, 10, and 13 dimensions for the Infant, Toddler (14 and 24m), and Pre-school (36 and 54m) surveys respectively, rather than the four dimensions examined in the simple MIRT. This model estimated the full covariance matrix among these sub-dimensions. In contrast, the fourth model addresses the subscale structure by adding latent variables for each of the primary subscales (READ, PIDA, PVR, and ALM) in addition to the component dimensions in effect, adding two levels of latent factors to the item structure. This approach is like the approaches used in the bi-factor model and the testlet model, [94] to account for the associations among the items within a subscale. This multivariate bi-factor model only includes correlations among the primary subscales rather than the smaller components.

To determine the best fitting model, we examined the Akaike information criteria (AIC) for each of the models fit to each of the age-level scales. Using this criteria, the lower AIC value

**Table 1. Revised StimQ$_2$ subscales with components.**

| StimQ$_2$ Infant | StimQ$_2$ Toddler | StimQ Preschool |
|---|---|---|
| **READ Subscale** | | |
| Bookreading Quantity | Bookreading Quantity | Bookreading Quantity |
| Diversity Concepts/Content | Diversity Concepts/ Content | Diversity Concepts |
| | | Diversity Content |
| Bookreading Quality | Bookreading Quality | Bookreading Quality |
| **PIDA Subscale** | | |
| — | — | Emergent Literacy |
| — | — | Math and Spatial Orientation |
| **PVR Subscale** | | |
| Everyday Routines | Everyday Routines | Everyday Routines |
| Play and Pretend | Play and Pretend | Play and Imagination |
| — | — | Self- Regulation |
| **ALM Subscale** | | |
| First Infant Toys | Symbolic Play | Symbolic Play |
| Activity/Manipulative Toys | Art | Art |
| Imagination Toys | Adaptive/ Fine Motor | Adaptive/ Fine Motor |
| — | Language | Language/Concepts |
| — | Life Size | — |

Note: The original StimQ did not include components, with one exception: the ALM subscale of the original StimQ included components that are consistent with those listed for the StimQ$_2$. StimQ$_2$ core score is calculated by summing READ, PIDA, and PVR subscales. StimQ$_2$ total score is calculated by summing READ (Reading), PIDA (Teaching), PVR (Verbal Responsivity), and ALM (Learning Materials) subscales.

indicates better fit. The results appear in Table 2. For all age-level scales, the AIC for the 4-factor model was higher than that for the 1-factor model, indicating that the 1-factor model was preferred over the 4-factor model. However, the two models that incorporated the component information (labeled Full Multivariable and Multivariable-Bifactor in the table) have lower AICs than both the 1- and 4-factor models for all age levels, indicating there are some local associations among the items within a component (e.g., PIDA Math and Spatial Orientation) not explained by the lower dimensional models. These findings suggest that the components underlying the subscales provide additional valuable information regarding the cognitive home environment and that their inclusion improves overall model fit across all time points.

**Reliability and subscore analysis.** In addition to the multivariate IRT analysis to examine the latent structure of the $StimQ_2$ instrument, we examined the added value of reporting each of the subscores at each age-level using Haberman [95] and Sinharay's [96] subscore analysis procedures. The subscore analysis is based on the classical test theory true score model. The procedure assumes that each observed subscore, denoted $S_{ij}$ for individual $i$ and subscore $j$, is a measure of the true subscore $\tau_{ij}$ with some amount of error according to the model

$$S_{ij} = \tau_{ij} + \epsilon_{ij}.$$

The classical definition of the reliability of a measure is the correlation between two parallel measures of the true score, or equivalently the squared correlation between the observed subscore $S_{ij}$ and the true score $\tau_{ij}$. That is, the reliability of subscore $j$ is $r_j = cor(S_{ij}, \tau_{ij})^2$. Another way to think about this definition of reliability is that is a measure of how well the observed score $S_{ij}$ predicts $\tau_{ij}$, as it would be the R-squared in a regression of $\tau_{ij}$ on $S_{ij}$. Haberman [95] describes this as the proportion reduction in mean squared error (PRMSE) for predicting $\tau_{ij}$ with the linear predictor $a_j + b_j S_{ij}$, where $a_j$ and $b_j$ are found by minimizing the mean squared error $MSE_j = E[(\tau_{ij} - a_j - b_j S_{ij})^2]$; essentially this is least squares regression for predicting the unobservable true score $\tau_{ij}$. With this setup we have

$$r_j = \frac{MSE_j}{Var(\tau_{ij})} \equiv PRMSE_j.$$

That is, the PRMSE of a subscale is the same as its reliability. The idea behind Haberman [95] and Sinharay's [96] analyses to examine the value added of subscores is to recognize that we can predict the true subscore $\tau_{ij}$ (e.g., READ true score) with the observed total score on the $StimQ_2$, which is $S_{i+} = \sum_j S_{ij}$, just as we can predict it with the observed score $S_{ij}$ on the subscale itself. Haberman [95] argues that if the total score (e.g., the total StimQ score) is a better predictor of the true score $\tau_{ij}$ than the observed subscore $S_{ij}$ in terms of the PRMSE, then the subscore adds no value beyond the total score.

**Table 2. Multi-dimensional item response theory models (MIRT) showing internal structure of the $StimQ_2$.**

|  | One Factor | Four Factors | Full Multivariable | Multivariable-Bifactor |
|---|---|---|---|---|
| Infant | 25178 | 25265 | 24422 | 24375 |
| Toddler 14 | 25351 | 25486 | 24709 | 24569 |
| Toddler 24 | 30391 | 30676 | 29832 | 29669 |
| Pre-school 36 | 22582 | 23272 | 22523 | 22004 |
| Pre-school 54 | 23706 | 23979 | 23407 | 22611 |

Note: Model fit assessed using Akaike information criteria (AIC)

**Table 3. Subscore analysis for the four subscales of the StimQ2.** Quantities are the PRMSE of the best linear predictor based on either the Subscore (Subscale Score) or the StimQ2 Total Score.

| | PVR Subscore Reliability | PVR StimQ$_2$ | READ Subscore Reliability | READ StimQ$_2$ | PIDA Subscore Reliability | PIDA StimQ$_2$ | ALM Subscore Reliability | ALM StimQ$_2$ |
|---|---|---|---|---|---|---|---|---|
| Infant | .719 | .518 | .690 | .535 | .377 | .758 | .735 | .508 |
| Toddler 14m | .707 | .590 | .725 | .561 | .536 | .672 | .802 | .677 |
| Toddler 24m | .690 | .675 | .706 | .686 | .535 | .644 | .786 | .768 |
| Preschool 36m | .652 | .676 | .753 | .700 | .690 | .632 | .800 | .771 |
| Preschool 54m | .790 | .608 | .753 | .640 | .732 | .476 | .807 | .706 |

Note: PVR = Verbal Responsivity, READ = Reading, PIDA = Teaching, ALM = Learning Materials

The results of the subscore analysis appear in Table 3. For each subscale of the StimQ$_2$ and each age group two numbers are reported. The reliability/PRMSE based on the subscore (Subscore subcolumn) and the PRMSE based on the StimQ$_2$ total score (StimQ$_2$ subcolumn); as discussed earlier, the subscore PRMSE corresponds to the reliability of the subscore (estimated by Cronbach's alpha).

For all ages and all subscores, except for the PIDA subscore on the Infant and Toddler scales and the PVR subscale on the Toddler scale at 36 months, the subscore PRMSEs are lower than the StimQ$_2$ PRMSEs (though this difference is small for the Toddler PVR at 36 months). This suggests that there is added value in reporting the subscales in addition to the StimQ$_2$ total score.

## Results for aim 2: Concurrent validity

Analyses for Aim 2 were performed using IBM SPSS Statistics version 28 [97]. Subgroup analyses of participants completing the StimQ2 in English vs. Spanish revealed similar patterns of associations. The majority of effect sizes were comparable across all ages, although we were not powered to show significance given the small number of mothers completing the StimQ2 in English. We therefore collapsed across language in all of our analyses. For all analyses, we report p-values corrected for multiple comparisons ($\alpha$ = .05 two-tailed) [98].

**StimQ$_2$ infant.** Results (see Table 4) revealed that all scales, subscales, and components of the StimQ$_2$ Infant (StimQ$_2$ I) had robust significant associations with early child language/ communication as measured by the CSBS-DP. StimQ$_2$ Infant Core and Total were also positively associated with measures of child temperament including regularity and intensity, with all subscales associated with infant regularity and READ associated with infant intensity.

**StimQ$_2$ toddler.** At child age 14 months (see Table 5), StimQ$_2$ Toddler (StimQ$_2$ T) Core and StimQ$_2$ T Total scores were significantly and positively associated with child cognitive ability, expressive and receptive language, and social emotional skills including imitation and attention. At this time point, associations with child cognitive ability were most pronounced for the PVR PIDA, and ALM scales, with a marginal association found between this domain of early development and the READ scale. Associations with expressive and receptive language outcomes were consistent for all subscales and components. READ (with the exception of the Diversity of Concepts component), PIDA, ALM, and PVR were all positively associated with child imitation and attention outcomes. None of the scales at this time point were related to child separation distress.

**Table 4. Concurrent relationships between StimQ₂ subscales and sub-components and developmental outcomes at 6 months.**

| | ALM | READ | | | | PIDA | PVR | | | StimQ | |
|---|---|---|---|---|---|---|---|---|---|---|---|
| | | **Quantity** | **Diversity** | **Quality** | **Total** | **Total** | **Routines** | **Pretend** | **Total** | **Core** | **Total** |
| Language (CSBS) | .21*** | .15** | .23*** | .23*** | .22*** | .25*** | .35*** | .28*** | .37*** | .37*** | .37*** |
| Social-Emotional (SITQ) | | | | | | | | | | | |
| Regularity | .16** | .16** | .11* | .17*** | .18*** | .12* | .17*** | .14** | .18*** | .22*** | .22*** |
| Intensity | 0.09$^t$ | 0.08 | .11* | 0.09$^t$ | .10$^t$ | 0.07 | 0.07 | 0.9$^t$ | 0.9$^t$ | .12$^t$ | .13$^t$ |
| Activity | -0.04 | -0.02 | -.04 | 0.01 | -0.02 | -0.06 | 0.01 | -0.06 | -0.02 | -0.03 | -0.04 |
| Soothability | 0.005 | -0.03 | .03 | 0.01 | -0.01 | 0.02 | -0.03 | -0.01 | -0.02 | -0.01 | -0.01 |

Note:

*$p < .05$;

**$p < .01$;

***$p < .001$;

$^t p < .09$;

PVR = Verbal Responsivity, READ = Reading, PIDA = Teaching, ALM = Learning Materials; CSBS = Communication and Symbolic Behavior Scales; SITQ = Revised Infant Temperament Questionnaire Short Form

Concurrent validity findings for StimQ₂ Toddler at child age 24 months (Table 6) were largely similar to those seen at 14 months. StimQ₂ T Core and StimQ₂ T Total were all significantly associated with greater performance in assessed cognitive, language, and social-emotional domains. Most subscales and components were also positively related to these outcomes; however, at this time point, READ Quality and READ Quantity were related to receptive language but not READ diversity of concepts or PIDA scores.

**Table 5. Concurrent relationships between StimQ₂ subscales and sub-components and developmental outcomes at 14 months.**

| | ALM | READ | | | | PIDA | PVR | | | StimQ | |
|---|---|---|---|---|---|---|---|---|---|---|---|
| | | **Quantity** | **Diversity** | **Quality** | **Total** | **Total** | **Routines** | **Pretend** | **Total** | **Core** | **Total** |
| Cognitive (Bayley) | .14* | .08 | 0.14* | .07 | .11$^t$ | .13* | .11$^t$ | .12$^t$ | .13* | .15* | .16* |
| Language (PLS-4) | | | | | | | | | | | |
| Expressive | .15** | .14* | .15** | .20*** | .20** | .18** | .27*** | .16** | .26*** | .28*** | .27*** |
| Receptive | 0.12* | .16** | .19** | .17** | .21*** | .27*** | .23*** | .19** | .24*** | .29*** | .28*** |
| Total | .16** | .17** | .20*** | .22*** | .24*** | .28*** | .30*** | .21*** | .30*** | .34*** | .33*** |
| Social-Emotional (ITSEA) | | | | | | | | | | | |
| Imitation | .25*** | 0.13* | 0.09 | .25*** | .19** | .27*** | .27*** | .27*** | .31*** | .33*** | .34*** |
| Attention | 0.13* | .34*** | 0.11$^t$ | .22** | .32*** | .13* | .26*** | .24*** | .29*** | .34*** | .32*** |
| Separation Distress | -0.11 | -0.06 | 0.07 | -0.12 | -0.06 | -0.01 | -0.09 | -.15 | -.13 | -0.10 | -0.11 |

Note:

*$p < .05$;

**$p < .01$;

***$p < .001$;

$^t p < .09$;

Note: PVR = Verbal Responsivity, READ = Reading, PIDA = Teaching, ALM = Learning Materials; PLS-4 = Preschool Language Scale, 4$^{th}$ edition; ITSEA = Infant-Toddler Social and Emotional Assessment-Revised

**Table 6. Concurrent relationships between StimQ$_2$ subscales and sub-components and developmental outcomes at 24 months.**

| | StimQ$_2$ Toddler 24 Months | | | | | | | | | | |
|---|---|---|---|---|---|---|---|---|---|---|---|
| | ALM | READ | | | | PIDA | PVR | | | StimQ | |
| | | Quantity | Diversity | Quality | Total | Total | Routines | Pretend | Total | Core | Total |
| Cognitive (Bayley) | .13* | .21*** | 0.10$^t$ | .15* | .22*** | .13* | .17** | .15** | .18** | .23*** | .23*** |
| Language (PLS-4) | | | | | | | | | | | |
| Expressive | .13* | 0.10 | 0.06 | .11$^t$ | 0.06 | 0.10 | .20*** | .10 | .19*** | .15* | .16* |
| Receptive | .15** | .15** | 0.10$^t$ | .14* | .18** | 0.09 | .22*** | .12* | .21*** | .21*** | .21*** |
| Total | .16** | .10$^t$ | 0.09$^t$ | .14* | .14* | .10$^t$ | .24*** | .12* | .22*** | .21*** | .21*** |
| Social-Emotional (ITSEA) | | | | | | | | | | | |
| Imitation | .29*** | .17** | .13* | .17** | .21*** | .13* | .36*** | .29*** | .38*** | .33*** | .35*** |
| Attention | .26*** | .31*** | .14** | .20*** | .32*** | .16* | .28*** | .24*** | .30*** | .34*** | .35*** |
| Separation Distress | -0.01 | 0.01 | -0.02 | 0.05 | 0.02 | -0.03 | 0.00 | 0.02 | 0.01 | 0.00 | 0.00 |

Note:

*$p < .05$;

**$p < .01$;

***$p < .001$;

$^t$p < .09;

Note: PVR = Verbal Responsivity, READ = Reading, PIDA = Teaching, ALM = Learning Materials; PLS-4 = Preschool Language Scale, 4$^{th}$ edition; ITSEA = Infant-Toddler Social and Emotional Assessment-Revised


**StimQ$_2$ preschool.** At child age 36 months (see Table 7), StimQ$_2$ Preschool Core (StimQ$_2$ P) and StimQ$_2$ P Total scores were significantly related to increased scores on child cognitive ability as measured using the Bayley. ALM and READ (except for Diversity of Content) were each associated with increased cognitive ability, PVR was marginally associated, while PIDA was not. The majority of subscales and components (except for in some cases READ Diversity of Concepts/Content, PIDA Literature, and PVR regulation) were significantly and positively related to increased expressive and receptive language abilities and social skills, and significantly and negatively related to externalizing behaviors including aggression and hyperactivity. Higher scores on ALM and PVR subscales were also marginally related to decreased reported attention problems.

While at 54 months, different measures were used to assess cognition, language, and social-emotional outcomes than used at 36 months, relationships between StimQ$_2$ Preschool and these early developmental domains were also observed (see Table 8). Regarding associations with cognition, at child age 54 months, while StimQ$_2$ P Core and StimQ$_2$ P Total scores were not associated, two of the PIDA subscale scores were significantly associated with child visual matching ability. StimQ$_2$ P Core and StimQ$_2$ P Total scores as well as all subscale and component scores were marginally or significantly positively related to non-verbal working memory. Furthermore, StimQ$_2$ P Core and StimQ$_2$ P Total scores as well as ALM, PIDA, and PVR subscale scores were significantly and positively correlated with child expressive and receptive language. READ was also significantly and positively correlated with expressive language ability at this age, but a relationship between the READ subscale, or diversity of reading and reading quality components was not established in this sample at this age. Finally, StimQ$_2$ P Core and StimQ$_2$ P Total scores were also significantly or marginally related to nearly all measures of child social-emotional outcome as measured using the BASC; in particular, they were positively related to social skills, and negatively related to attention problems, and externalizing behaviors including hyperactivity and aggression. ALM, READ, PIDA subscale scores had

**Table 7. Concurrent relationships between StimQ$_2$ subscales and sub-components and developmental outcomes at 36 months.**

| | ALM | READ | | | | | PIDA | | | PVR | | | StimQ | |
|---|---|---|---|---|---|---|---|---|---|---|---|---|---|---|
| | | Quantity | Diversity Concepts | Diversity Content | Quality | Total | Lit | Math | Total | Routines | Regulation | Total | Core | Total |
| **StimQ Preschool (36 months)** | | | | | | | | | | | | | | |
| Cognitive (Bayley) | .16* | .12$^t$ | .15* | 0.08 | .17* | .16* | 0.05 | 0.07 | 0.07 | .11$^t$ | 0.10 | .12$^t$ | .15* | .16* |
| Language | | | | | | | | | | | | | | |
| WCJ Picture Vocab | .30*** | .25*** | .21*** | .22*** | .22*** | .29*** | .13* | .16** | .17** | .22*** | .22*** | .25*** | .30*** | .33*** |
| CELF Expressive | .26*** | .14* | 0.08 | .14* | .25*** | .19** | .11$^t$ | .19** | .18** | .24*** | .12* | .23*** | .26*** | .24*** |
| CELF Receptive | .18** | .11$^t$ | .13* | 0.09 | .20** | .16* | 0.04 | .14* | .11$^t$ | .24*** | .18** | .25*** | .22*** | .21*** |
| CELF Core Language | .22*** | .11$^t$ | 0.10$^t$ | .12$^t$ | .24*** | .17** | .11$^t$ | .16** | .16** | .27*** | .15* | .26*** | .24*** | .25*** |
| Social-Emotional (BASC) | | | | | | | | | | | | | | |
| Attention Problems | -.16$^t$ | -0.03 | -0.01 | -0.02 | -0.02 | -0.03 | -0.06 | -.11 | -.10 | -.15$^t$ | -0.09 | -.14$^t$ | -.11$^t$ | -.13 |
| Social Skills | .29*** | .15* | .17** | .15** | .23*** | .21*** | .15** | .28*** | .25*** | .24*** | 0.28*** | .29*** | .30*** | .32*** |
| Hyperactivity | -.16* | -.13* | -.15* | -0.10 | -.12* | -.16* | -0.05 | -.10 | -0.09 | -.15* | -.14* | -.17* | -.18* | -.18* |
| Aggression | -.22*** | -.14* | -.19** | -0.05 | -.14* | -.16* | -0.11$^t$ | -.13$^t$ | -.13* | -.16** | -.14* | -.18** | -.20*** | -.22** |
| Externalizing | -.21*** | -.15* | -.19** | -0.09 | -.15* | -.18** | -0.08 | -.12* | -.12$^t$ | -.18** | -.16** | -.19** | -.21*** | -.22*** |

Note:

*$p < .05$;

**$p < .01$;

***$p < .001$;

$^t p < .09$;

Note: PVR = Verbal Responsivity, READ = Reading, PIDA = Teaching, ALM = Learning Materials; WCJ = Woodcock Johnson; CELF = The Clinical Evaluation of Language Fundamentals

similar associations with these outcomes; however, the PVR subscale was significantly related to increased social skills and decreased attention problems and aggression, but only the PVR regulation component of this subscale was associated with reduced hyperactivity.

## Discussion

The results from the current study provide support for the use of the StimQ$_2$ as a psychometrically valid and reliable instrument for assessing cognitive stimulation in the home from infancy through the preschool years. Analysis of each version of the instrument, StimQ$_2$-I, StimQ$_2$-T, and StimQ$_2$-P showed adequate internal consistency for both the overall measures and for the majority of the subscales. Furthermore, StimQ$_2$-I, StimQ$_2$-T, and StimQ$_2$-P all had moderate to strong associations with commonly used measures of child development at each of the respective time points (CSBS DP and SITQ in infancy; Bayley, PLS-4, and ITSEA in toddlerhood; and Bayley, Woodcock- Johnson/Woodcock-Munoz, CELF-2, and EOWPVT/ ROWPVT in the preschool years), demonstrating its use as a valid correlate of important child development outcomes over this time period.

These findings build on prior investigations of the original StimQ, which demonstrated the measure to have good reliability and validity [27]. The StimQ$_2$ maintains, and in some cases improves, these psychometric properties, but also offers other distinct advantages for usability.

**Table 8. Concurrent relationships between StimQ₂ subscales and sub-components and developmental outcomes at 54 months.**

| | | StimQ Preschool (54 months) | | | | | | | | | | | | | |
| | ALM | READ | | | | | PIDA | | | PVR | | | | StimQ | |
| | | Quantity | Diversity Concepts | Diversity Content | Quality | Total | Literacy | Math | Total | Routines | Pretend | Regulation | Total | Core | Total |
| --- | --- | --- | --- | --- | --- | --- | --- | --- | --- | --- | --- | --- | --- | --- | --- |
| **Cognitive (WCJ)** | | | | | | | | | | | | | | | |
| Visual Matching | 0.08 | 0.06 | -0.04 | 0.06 | 0.00 | 0.07 | .24* | .14 | .20* | 0.08 | 0.00 | 0.11 | 0.08 | 0.12 | 0.12 |
| Working Memory | .18** | .25*** | .13* | .21*** | .14* | .23*** | .13* | .11[t] | .18* | .18** | .12[t] | .15* | .18** | .24*** | .24*** |
| **Language** | | | | | | | | | | | | | | | |
| EOWPVT | .26*** | .15* | 0.01 | .12[t] | 0.07 | .12[t] | .17** | .21*** | .22*** | .31*** | .26** | .17** | .31*** | .27*** | .29*** |
| ROWPVT | 0.08 | .12[t] | 0.02 | 0.04 | 0.07 | 0.09 | .13[t] | .18** | .18* | .29*** | .17* | .13* | .26*** | .22*** | .21** |
| **Social-Emotional (BASC)** | | | | | | | | | | | | | | | |
| Attention Problems | -.16** | -.20** | -.14* | -.15* | -.18** | -.21*** | -.19** | -.18** | -.21*** | -.23*** | -0.09 | -.24*** | -.23*** | -.28*** | -.27*** |
| Social Skills | .16** | .20** | .19** | .16** | .26*** | .24*** | .18** | .18** | .20** | .22*** | .18** | .25*** | .26*** | .30*** | .30*** |
| Hyperactivity | -.12[t] | -.14[t] | -.12[t] | -.11 | -.17* | -.17* | -.13[t] | -0.09 | -.12[t] | -0.09 | -0.01 | -.16* | -0.08 | -.16* | -.16* |
| Aggression | -0.11[t] | -.15* | -.15* | -0.09 | -.15* | -.16** | -.21*** | -.14* | -.20** | -.13* | -0.08 | -.21*** | -.16** | -.22*** | -.21** |
| Externalizing | -.13* | -.16* | -0.15* | -.11[t] | -.18** | -.19** | -.18** | -.13* | -.17** | -.12[t] | -0.04 | -.20** | -.13* | -.21** | -.21** |

Note:

*$p < .05$;

**$p < .01$;

***$p < .001$;

[t] $p < .09$;

Note: PVR = Verbal Responsivity, READ = Reading, PIDA = Teaching, ALM = Learning Materials; WCJ = Woodcock Johnson; EOWPVT = Expressive One Word Picture Vocabulary Test;
ROWPVT = Receptive One Word Picture Vocabulary Test

The StimQ$_2$ provides a version of this instrument that has simultaneously removed outdated items due to changes over time in technology/learning materials for children and also added new items that capture domains of cognitive stimulation that have been more recently evidenced to play a role in supporting early developmental outcomes. For example, the StimQ$_2$ has expanded the READ and PVR subscales to include items that further capture aspects of the home literacy environment and broad language input to children. For instance, the READ scale for infants and toddlers now includes additional questions about the quality of parent-child bookreading interactions, such as talking about the emotions and mental states of characters in books. The PVR scale for infants and toddlers was also expanded to include additional questions regarding talking to children during everyday routines, such as while performing housework or chores. Scales for these two age ranges also now include items pertaining to additional forms of storytelling outside bookreading, such as folktales, oral story telling, or personal narratives, which may be particularly relevant for families from diverse cultural backgrounds or those with lower education and literacy [69–71]. The Preschool form of the StimQ$_2$ also greatly added to the PVR subscale by including a Pretend Play component, which aims to capture information about how symbolic pretend play is supported in the home. This is important because research on guided play, in which adults scaffold children's without infringing on their autonomy has been shown to relate to child curiosity, motivation, and learning [99–101].

The StimQ$_2$ also offers the advantage relative to its predecessor of being structured so that each subscale is comprised of distinct components (e.g., StimQ$_2$-P READ subscale includes 3 components: Read Quantity, Read Quality, and Read Diversity of Concepts) that can be assessed either collectively or independently. This change in the new StimQ forms gives greater flexibility to users who would like to select key aspects of cognitive stimulation to target in their assessment. Furthermore, the ability to select a smaller subset of components of cognitive stimulation to assess provides the possibility of an even shorter administration time, further lessening this burden relative to other existing measures. In addition, this updated version also allows users to select from two different scoring systems when considering a total score including 1) one with all four subscale scores combined together (StimQ$_2$-Total), and 2) a "core" cognitive stimulation score combining the scores from the READ, PIDA, and PVR subscales (StimQ$_2$-Core) but excluding ALM. The current findings suggest that the Total and Core Scores are similarly associated with children's outcomes across time. However, prior research investigating the cognitive home environment with the HOME included measurement of availability of learning materials, and revealed that children's access to books, toys, and games is a strong predictor of child outcomes [102]. The current findings showing associations between the ALM scale and children's development complement that prior work. Given the longer length of administration for the ALM, the StimQ$_2$ now provides users with flexibility in deciding whether to include availability of learning materials when determining how to score the instrument.

It should be noted with caution that certain StimQ$_2$ subscales at specific ages (e.g., Infant and Toddler PIDA and Preschool PVR at 36m) did not meet optimal standards for internal consistency/reliability. It is possible that in some cases (e.g., PIDA), this is due to the small number of items comprising these subscales. It is possible then that less information is garnered from these small number of items alone than can obtained by the full StimQ$_2$ survey itself. Users of the StimQ$_2$ should consider using multiple means to assess these constructs if planning to use these specific subscales of the StimQ$_2$ independently from the rest of the scale. We also note that some of the components of the READ scale, primarily those related to the diversity of concepts and contents of storybooks, were not robustly associated with child developmental outcomes. Nevertheless, these items were maintained as they are conceptually useful for providing information about the early home literacy environment and do not lessen the

overall usefulness of the READ subscale in predicting school readiness outcomes as evidenced by overall concurrent validity findings.

This study also provides additional support for extending use of the StimQ$_2$ with preschool age children [31]. This is important as 1) research demonstrates that cognitive stimulation in the home during the preschool years continues to be an important predictor of school readiness outcomes [29, 30], and 2) use of the StimQ over the past decade has made it evident that there is great need during the preschool period for researchers and practitioners to capture this information about cognitive stimulation in the home to understand how best to tailor intervention efforts [36, 103, 104].

Evidence in support of the StimQ$_2$ as a reliable and valid instrument for assessing the cognitive home environment from infancy through the preschool period is important because, like the original version, the StimQ$_2$ has numerous benefits for administration and interpretation in comparison to existing measures. Firstly, it offers much in the way of practicality given that it is relatively short to administer, easy to train staff to administer, is available as a free download, and is available for use in multiple languages. It has also been used to a large degree in research (at least 93 publications/studies in and outside of our lab), demonstrating both feasibility of using this instrument with many populations and the great need that exists for such a measure. Beyond issues related to ease and feasibility of administration, the StimQ$_2$ also offers advantages by design in that: 1) it is structured to reduce social desirability bias; and critically 2) it simultaneously measures multiple aspects of early cognitive stimulation in the home.

Another advantage of the StimQ$_2$ is that it has now been shown to be reliable and valid in Latinx families with low SES. This is critically important given the lack of research on measures of cognitive stimulation for this population, whom are the largest ethnic minority group in the US. In addition, because poverty-related disparities in early childhood development and school readiness have been deemed a public health crisis [14–16], and early cognitive stimulation in the home is one powerful and potentially modifiable factor in enhancing outcomes for children from low-SES homes [23–26], an easy-to-administer instrument to assess cognitive home environment is particularly useful for investigators who work with these groups. The StimQ$_2$ format may be particularly conducive for work involving families for whom access to the home may be an issue. Also, data collection by interview helps to alleviate potential concerns related to parental literacy sometimes encountered in parent-report measures administered to populations with low levels of formal education.

Findings from the current study, while primarily aiming to demonstrate validity and reliability of the StimQ$_2$ instrument, also notably contribute to the literature on the role of cognitive stimulation in the home for early child development across a range of domains. The concurrent associations found during the infant, toddler, and preschool period between reported cognitive stimulation in the home and aspects of child social, language, and cognitive abilities support similar associations previously documented [4, 5, 47, 105], and extend these findings to a primarily Hispanic/Latinx immigrant population with low income. This is important given the impact of poverty on child development and given challenges more generally to replication in developmental science. Altogether, this provides a strong rationale and evidence for the use of the StimQ$_2$ in assessing parents' cognitive stimulation in the home.

## Limitations and future directions

While this study has many strengths, it also has several limitations. Given that Hispanic/Latinx families are a heterogeneous and diverse group with large representation in the United States,

we viewed the inclusion of a largely Hispanic/Latinx, low-income, immigrant population as a strength in our study. However, we recognize that these findings may not generalize to other populations. Nevertheless, as previously noted, the StimQ has been widely used with other populations diverse in race, ethnicity, and SES both in the United States and internationally [43, 106–108]. In addition, as part of ongoing efforts, new data are being collected that will enable the investigation of whether the StimQ$_2$ has broad applicability across various cultures and a range of SES groups. This includes current work with several samples that include larger numbers of Black, White, and multi-racial families [37, 109, 110].

In addition, we note that the timing of the data collection for this study (between 2005 and 2012) may be viewed as a limitation because it took place prior to changes in access and use of digital devices among children and families. However, the StimQ$_2$ was not designed to assess cognitive stimulation in the context of screen time and digital media, and the key domains that are assessed have not otherwise undergone significant changes. Notably, our analyses of concurrent validity assessed StimQ$_2$ in relation to other important instruments that are still valid today, speaking to the relevance of the measure in its current form. We also recognize that while we investigated associations between StimQ$_2$ and a range of social, language, and cognitive outcomes for children, we were not able to measure and consider all aspects of early child development that may relate to success in formal education such as pre-literacy and numeracy skills; future research conducted with this instrument should aim to take a broader range of skills into account when possible. In addition, while data collection by interview has been viewed as a strength in working with low-SES and low-literacy populations, we acknowledge that this also places some burden on training staff for data collection, albeit lower than that associated with observational methods.

StimQ to date has largely been used as a tool by researchers; however, it may also have practical applicability in clinical settings for helping to determine how and when to intervene with families (and whether intervention is successful). Thus, we are also engaged in efforts to broaden the use and optimize efficiency of this measure while preserving its unique methods of reducing bias. For example, ongoing work aims to validate the use of StimQ$_2$ in a purely self-report format so that researchers and practitioners may select this methodology if it is deemed most suitable for their purpose/population.

## Conclusions

In sum, this study provides evidence of the StimQ$_2$ as a valid and reliable tool for broadly evaluating cognitive stimulation in the home for children from infancy through the preschool period in a low-cost and efficient manner. This instrument offers significant advantages for assessment of cognitive stimulation including its comprehensive assessment of multiple attributes of cognitive stimulation, its use of interview techniques designed to reduce self-report bias, the possibility for flexible use of subscales and components, and its suitability for evaluating cognitive stimulation in homes of low-SES populations.

## Supporting information

**S1 Appendix. Possible score ranges for the StimQ$_2$.**
(PDF)

**S1 Table. Number of items comprising each subscale of the original StimQ and StimQ$_2$.**
(PDF)

## Acknowledgments

We would like to thank the many individuals who contributed to this project, including Jenny Arevalo, Samantha Berkule, Catherine Tamis-LeMonda, Lisa White, and Kristina Vlahovi-cova. We also thank all of the children and parents who participated in this research.

## Author Contributions

**Conceptualization:** Carolyn Brockmeyer Cates, Erin Roby, Caitlin F. Canfield, Matthew Johnson, Caroline Raak, Adriana Weisleder, Benard P. Dreyer, Alan L. Mendelsohn.

**Data curation:** Carolyn Brockmeyer Cates, Erin Roby, Caitlin F. Canfield, Matthew Johnson, Caroline Raak, Adriana Weisleder, Benard P. Dreyer, Alan L. Mendelsohn.

**Formal analysis:** Carolyn Brockmeyer Cates, Erin Roby, Caitlin F. Canfield, Matthew Johnson, Adriana Weisleder, Benard P. Dreyer, Alan L. Mendelsohn.

**Funding acquisition:** Alan L. Mendelsohn.

**Investigation:** Carolyn Brockmeyer Cates, Erin Roby, Caitlin F. Canfield, Adriana Weisleder, Benard P. Dreyer, Alan L. Mendelsohn.

**Methodology:** Carolyn Brockmeyer Cates, Erin Roby, Caitlin F. Canfield, Caroline Raak, Adriana Weisleder, Benard P. Dreyer, Alan L. Mendelsohn.

**Project administration:** Carolyn Brockmeyer Cates, Erin Roby, Caroline Raak, Adriana Weisleder, Alan L. Mendelsohn.

**Resources:** Alan L. Mendelsohn.

**Supervision:** Carolyn Brockmeyer Cates, Adriana Weisleder, Alan L. Mendelsohn.

**Validation:** Carolyn Brockmeyer Cates, Erin Roby, Caitlin F. Canfield, Caroline Raak, Adriana Weisleder, Benard P. Dreyer, Alan L. Mendelsohn.

**Visualization:** Carolyn Brockmeyer Cates, Erin Roby.

**Writing – original draft:** Carolyn Brockmeyer Cates, Erin Roby, Caitlin F. Canfield, Matthew Johnson, Caroline Raak, Adriana Weisleder, Benard P. Dreyer, Alan L. Mendelsohn.

**Writing – review & editing:** Carolyn Brockmeyer Cates, Erin Roby, Caitlin F. Canfield, Matthew Johnson, Caroline Raak, Adriana Weisleder, Benard P. Dreyer, Alan L. Mendelsohn.

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
