## [Decision Letter · Decision Letter 0]

14 Feb 2023

PONE-D-23-00912Validation of the StimQ2:

A parent-report measure of cognitive stimulation in the homePLOS ONE

Dear Dr. Roby,

Thank you for submitting your manuscript to PLOS ONE. After careful consideration, we feel that it has merit but does not fully meet PLOS ONE’s publication criteria as it currently stands. Therefore, we invite you to submit a revised version of the manuscript that addresses the points raised during the review process.

We look forward to receiving your revised manuscript.

Kind regards,

Lu Hua Chen, PhD, M.D.

Academic Editor

PLOS ONE

Reviewers' comments:

Reviewer's Responses to Questions

**Comments to the Author**

1. Is the manuscript technically sound, and do the data support the conclusions?

Reviewer #1: Yes

Reviewer #2: Yes

2. Has the statistical analysis been performed appropriately and rigorously? 

Reviewer #1: Yes

Reviewer #2: I Don't Know

3. Have the authors made all data underlying the findings in their manuscript fully available?

Reviewer #1: Yes

Reviewer #2: No

4. Is the manuscript presented in an intelligible fashion and written in standard English?

Reviewer #1: Yes

Reviewer #2: No

5. Review Comments to the Author

Reviewer #1: Thank you very much for the opportunity to review this well-written paper. I found the introduction and descriptions of complex statistics particularly well done. However, I have a few minor suggestions and comments which I believe would further improve the manuscript:

You already mention in intro that the StimQ is available in English & Spanish, and specify other languages - repetition between lines 108 and 206

It would be useful to know a bit more information about the participants - what percentage of those from the trial were included in this analysis? Were they the same participants at each time, accounting for drop outs over time?

What statistical programme was used for data analysis?

The sentence on missing data in the Stim Q section should be in the data analytic plan. In addition, it would be useful to know the percentages of missing data at each time point.

Note indicating what asterisks or other superscript symbols stand for is missing in tables 4a/4b/4c/4d?

Was there any PPI involvement in developing the new version of this questionnaire? This can be helpful in ensuring relevance of items and reducing response errors. If not, this could be included in the limitations section.

Reviewer #2: The need for an instrument like this to minimized the need for home observation is important and understood. The use of standard surveys to conduct criterion validation is appreciated. Here are some concerns that I would appreciate being addresssed:

1. The sample is essentially Hispanic with very low level of education so how can these results be at all generalized to a different demographic? Also were there differences in results between those doing the study in Spanish vs English?

2. The tables do not stand alone; abbreviations aren't exaplained, possible scores for each survey and each subscale aren't provided.

3. The correlations are significant but very small? Since there are so many of them was some sort of correction factor, eg. Bonferroni applied?

4. I was not able to follow your statistical rationale at all. This wasn't explained in plain English.

5. How were the original data gathered? I just need some basics without having to go to another article to find out.

6. Were any partial correlations used/needed? What about controlling for marital status or some other demographic that may make a difference in the correlation? Were the 546 mothers that homogeneous within each child age group?

7. Data is a plural word so it should be Data were....

6. PLOS authors have the option to publish the peer review history of their article (what does this mean?). If published, this will include your full peer review and any attached files.

Reviewer #1: **Yes: **Rebecca Appleton

Reviewer #2: No

---

## [Author Response · Author response to Decision Letter 0]

12 May 2023

Response to Reviewers

Reviewer #1

1. You already mention in intro that the StimQ is available in English & Spanish, and specify other languages - repetition between lines 108 and 206.

Thank you for drawing this to our attention. We have removed the redundant information from the section titled, “StimQ” (Page 9)

2. It would be useful to know a bit more information about the participants - what percentage of those from the trial were included in this analysis? Were they the same participants at each time, accounting for drop outs over time?

We have added the total number of participants from larger trial that were enrolled at each time point and the percentage included in this analysis to the 

Participants section (Page 12). The participants were not necessarily all the same at each time point. Some families left and returned to the trial (e.g., skipped an assessment, moved out of the country and came back, etc.). Therefore, not all differences in the percentages available at each time point were due to dropout. We now clarify this by stating, “Mother-child dyads who were assessed on at least one of five possible time points (child age 6, 14, 24, 36, and 54 months) and had data for the variables of interest were included in analyses. Mothers did not have to maintain participation across all time points to be included.”

3. What statistical programme was used for data analysis?

We have added information regarding the statistical programs and packages used in our data analysis. This includes R (mirt package) for analyses related to Aim 1 (Page 18) and SPSS for analyses related to Aim 2 (Page 24).

4. The sentence on missing data in the Stim Q section should be in the data analytic plan. In addition, it would be useful to know the percentages of missing data at each time point.

We have moved the details regarding missing data from the StimQ section to the Data Analytic Plan section (Page 17). In addition, we now include information regarding the percentages of missing data at each time point in that same section. More specifically, we state, “Data was imputed for at least one item at the following rates across time points: 6 months (15.76%), 14 months (17.65%), 24 months (.53%), 36 months (5.30%), 54 months (.73%).”

5. Note indicating what asterisks or other superscript symbols stand for is missing in tables 4a/4b/4c/4d?

We have added notes indicating what asterisks and other superscript symbols stand for in Tables 4a-4d.

6. Was there any PPI involvement in developing the new version of this questionnaire? This can be helpful in ensuring relevance of items and reducing response errors. If not, this could be included in the limitations section.

Thank you for this question. We added information regarding how research participants were informally included in the process of developing the new version of the StimQ and how members of our team provided feedback regarding cultural appropriateness of the measure to the introduction (Page 11). 

Reviewer #2

1. The sample is essentially Hispanic with very low level of education so how can these results be at all generalized to a different demographic? Also, were there differences in results between those doing the study in Spanish vs English?

It is true that the sample in this study was primarily Hispanic/Latinx with low socioeconomic status. However, there group was fairly heterogeneous in terms of their racial identification, country or origin, and educational levels which we have added to the method section (Page 12).

We view the demographics of our sample as a strength and highlight this in the introduction and conclusion. Latinx families are the largest ethnic minority group in the United States, making up 19% of the population (62.1 million people; US Census Bureau, 2020), and are more likely to experience poverty. Our findings are therefore likely to be applicable to a large number of children and families. To further support the assertion that study of this population is needed, we have added the number of Hispanic/Latinx individuals living in the US to the introduction (Page 8). 

We also agree that generalization to other populations may be an issue. We acknowledge this as a limitation in the discussion. However, we have also added additional information regarding the wide use of the StimQ2 in other populations that are diverse in race, ethnicity, and SES. We also added a clearer description of the ongoing efforts we are engaged in to assess the applicability of the StimQ2 with other populations, including studies with samples that include larger numbers of Black, White, and multi-racial families (Page 37). 

Subgroup analyses of participants completing the StimQ2 in English vs. Spanish revealed similar patterns of associations. The majority of effect sizes for English and Spanish were comparable across all ages, although we were not powered to show significance given the small number of mothers completing the StimQ2 in English. We therefore collapsed across language in all of our analyses. We have added this information to the results section (Page 24).

2. The tables do not stand alone; abbreviations aren't explained, possible scores for each survey and each subscale aren't provided.

We have added notes that explain abbreviations to all tables. In addition, we created a supplementary Appendix that shows the possible scores for each of the StimQ2 surveys, including subscales, and components.

3. The correlations are significant but very small? Since there are so many of them was some sort of correction factor, eg. Bonferroni applied?

The correlations reported in the paper that are significant fall into the range of .10 to .38. These effect sizes are consistent with other studies examining concurrent validity of measures of cognitive stimulation and the home literacy environment (Bradley et al., 2001; Elardo et al., 1975; Evans et al., 2000). In the revision we now report p-values corrected for multiple comparisons (� = .05 two-tailed; Benjamini & Hochberg, 1995) (Page 24).

4. I was not able to follow your statistical rationale at all. This wasn't explained in plain English.

We have revised the statistical rationale throughout the Results section to provide additional clarity and improve readability. In addition, we added a figure (i.e., Figure 1) that depicts the 4 multi-dimensional item response theory (MIRT) models, which helps to demonstrate the differences in complexity between them. 

5. How were the original data gathered? I just need some basics without having to go to another article to find out.

We have included additional details regarding data collection for the original StimQ measure. This information appears at the end of the section called, “StimQ” (Page 9). 

We now state, “Data for the original StimQ were collected at a large, urban public hospital (a small percentage were collected at private pediatric clinics) through a series of studies focused on mothers of children between 12 to 36 months. Mothers were primarily Latinx or Black immigrants with low socioeconomic status. A smaller percentage were White with medium to high SES. The survey was administered to mothers by trained interviewers in their primary language during their pediatric visit.”

6. Were any partial correlations used/needed? What about controlling for marital status or some other demographic that may make a difference in the correlation? Were the 546 mothers that homogeneous within each child age group?

The factors proposed, such as marital status and other sociodemographic factors are likely to have strong associations with early child development outcomes, and those associations are mediated by the cognitive home environment (Brooks-Gunn et al., 2005; Garner et al., 2021). Therefore, we do not consider the use of such partial correlations as needed or indicated in the context of the analyses used in this study, which are not broad analyses related to constructs, but rather analyses related to instrument validity. In other words, we are interested in understanding the extent to which the instrument is able to reveal variation in (or covary with) these constructs rather than to make any claims about that how the parenting behaviors contribute to these outcomes. As such, we do not see a conceptual rationale for making these adjustments. This strategy is in alignment with prior validation studies of prominent instruments measuring the cognitive home environment (e.g., the HOME, one of the most commonly used instruments in the field), which have not controlled for demographic variables (e.g., Bradley & Caldwell, 1984; Elardo et al., 1975), or have instead considered relations between sociodemographic variables and the instrument separate from child outcomes (e.g., Bradley & Caldwell, 1984; Bradley & Caldwell, 1987; Bradley et al., 1988).

The 546 mothers were not homogeneous within each child age group. There were different numbers of mothers with available data at each of the time points. “We now clarify this by stating, “Mother-child dyads who were assessed on at least one of five possible time points (child age 6, 14, 24, 36, and 54 months) and had data for the variables of interest were included in analyses. Mothers did not have to maintain participation across all time points to be included.” 

7. Data is a plural word so it should be Data were....

We have replaced the two instances where data were referred to in the singular. 

References

Benjamini, Y., & Hochberg, Y. (1995). Controlling the false discovery rate: a practical and powerful approach to multiple testing. Journal of the Royal statistical society: series B (Methodological), 57(1), 289-300.

Bradley, R. H., & Caldwell, B. M. (1984). The HOME Inventory and family demographics. Developmental Psychology, 20(2), 315.

Bradley, R. H., & Caldwell, B. M. (1984). The relation of infants' home environments to achievement test performance in first grade: A follow-up study. Child development, 803-809.

Bradley, R. H., & Caldwell, B. M. (1979). Home observation for measurement of the environment: A revision of the preschool scale. American Journal of Mental Deficiency, 84(3), 235–244.

Bradley, R. H., Caldwell, B. M., Rock, S. L., Hamrick, H. M., & Harris, P. (1988). Home observation for measurement of the environment: Development of a home inventory for use with families having children 6 to 10 years old. Contemporary Educational Psychology, 13(1), 58-71.

Brooks-Gunn, J., & Markman, L. B. (2005). The contribution of parenting to ethnic and racial gaps in school readiness. The future of children, 139-168.

Elardo, R., Bradley, R., & Caldwell, B. M. (1975). The relation of infants' home environments to mental test performance from six to thirty-six months: A longitudinal analysis. Child development, 71-76.

Evans, Mary Ann, Deborah Shaw, and Michelle Bell. "Home literacy activities and their influence on early literacy skills." Canadian Journal of Experimental Psychology/Revue canadienne de psychologie expérimentale 54, no. 2 (2000): 65.

Garner, A., & Yogman, M. (2021). Committee on Psychosocial Aspects Of Child And Family Health Sodabp Council On Early Childhood. Preventing Childhood Toxic Stress: Partnering With Families and Communities to Promote Relational Health. Pediatrics, 148(2), e2021052582.

---

## [Editor Report · Decision Letter 1]

23 May 2023

Validation of the StimQ2:

A parent-report measure of cognitive stimulation in the home

PONE-D-23-00912R1

Dear Dr. Roby,

We’re pleased to inform you that your manuscript has been judged scientifically suitable for publication and will be formally accepted for publication once it meets all outstanding technical requirements.

Kind regards,

Lu Hua Chen, PhD, M.D.

Academic Editor

PLOS ONE

Additional Editor Comments (optional):

I am satisfied with the revision.
---

## [Editor Report · Acceptance letter]

14 Jul 2023

PONE-D-23-00912R1 

Validation of the StimQ2:
A parent-report measure of cognitive stimulation in the home 

Dear Dr. Roby:

I'm pleased to inform you that your manuscript has been deemed suitable for publication in PLOS ONE. Congratulations! Your manuscript is now with our production department. 

Kind regards, 

on behalf of

Dr. Lu Hua Chen 

Academic Editor

PLOS ONE